# Inhibitory Effectiveness in Delayed-Rectifier Potassium Current Caused by Vortioxetine, Known to Be a Novel Antidepressant

**DOI:** 10.3390/biomedicines10061318

**Published:** 2022-06-03

**Authors:** Hung-Tsung Hsiao, Jeffrey Chi-Fei Wang, Sheng-Nan Wu

**Affiliations:** 1Department of Anesthesiology, National Cheng Kung University Hospital, College of Medicine, National Cheng Kung University, Tainan City 70101, Taiwan; aneshsiao@gmail.com (H.-T.H.); cfwang0505@gmail.com (J.C.-F.W.); 2Department of Physiology, National Cheng Kung University Medical College, Tainan City 70101, Taiwan; 3Institute of Basic Medical Sciences, National Cheng Kung University Medical College, Tainan City 70101, Taiwan

**Keywords:** vortioxetine (1-(2-((2,4-dimethylphenyl)thio)phenyl)piperazine), voltage-gated K^+^ current, delayed-rectifier K^+^ current, *erg*-mediated K^+^ current, inactivation kinetics, binding scheme, cumulative inhibition, serotonin reuptake, pituitary cell, neuroblastoma cell

## Abstract

Vortioxetine (VOR) is recognized to exert antidepressant actions. However, whether this drug modifies ionic currents in excitable cells remains unclear. The aim of this study was to explore the electrophysiological effects of VOR and other related compounds in pituitary GH_3_ cells and in Neuro-2a cells. VOR suppressed the delayed-rectifier K^+^ current (*I*_K(DR)_) in a concentration-, time-, and state-dependent manner. Effective IC_50_ values needed to inhibit peak and sustained *I*_K(DR)_ were computed to be 31.2 and 8.5 μM, respectively, while the K_D_ value estimated from minimal binding scheme was 7.9 μM. Cell exposure to serotonin (10 μM) alone failed to alter *I*_K(DR)_, while fluoxetine (10 μM), a compound structurally similar to VOR, mildly suppressed current amplitude. In continued presence of VOR, neither further addition of propranolol nor risperidone reversed VOR-mediated inhibition of *I*_K(DR)_. Increasing VOR concentration not only depressed *I*_K(DR)_ conductance but also shifted toward the hyperpolarized potential. As the VOR concentration was raised, the recovery of *I*_K(DR)_ block became slowed. The *I*_K(DR)_ activated by a downsloping ramp was suppressed by its presence. The inhibition of *I*_K(DR)_ by a train pulse was enhanced during exposure to VOR. In Neuro-2a cells, this drug decreased *I*_K(DR)_. Overall, inhibitory effects of VOR on ionic currents might constitute another underlying mechanism of its actions.

## 1. Introduction

Vortioxetine (VOR, Brintellix^®^, Trintellix^®^, Lu AA21004, 1-(2-((2,4-dimethylphenyl)thio)phenyl)piperazine), an *N*-arylpiperazine in which the aryl group is specified as 2-[(2,4-dimethylphenyl)sulfanyl]phenyl, is widely regarded to be a novel oral antidepressant drug approved for the treatment of major depressive disorders in adults [1,2,3,4,5,6,7,8,9,10,11]. This drug is thought to be an inhibitor of serotonin (5-hydroxytryptamine, 5-HT) reuptake [12,13]. However, the underlying mechanism of its actions is not entirely clear, because its antidepressant or anxiolytic effects are believed to presumably be associated with the increased serotonin levels in extracellular milieu due to inhibition of serotonin reuptake into the cell interior and possibly with an interaction with certain receptors to which serotonin can bind [3,5,14,15].

Despite the inhibition of the serotonin reuptake transporter, it has been demonstrated that VOR might exert an additional action on glutamate neurotransmission [16]. Chronic treatment with VOR has been reported to improve the responsiveness to an acute stress acting through the ventral hippocampus in a glucocorticoid-dependent way [17]. It has also been proposed that this drug can facilitate the function of the hypothalamic-pituitary-adrenal axis [15,17]. This effect is thought to be mediated through its agonistic effect on serotonin receptors [3,15]. On the other hand, this compound has been reported to exert some cardiovascular adverse effects [18,19]. However, as far as we know, the ionic mechanism of VOR-induced actions through which it produces anti-depressant or anxiolytic actions still remains largely unanswered.

The voltage-gated K^+^ (K_V_) channels are regarded as having a great influence in determining membrane excitability associated with delayed-rectifier K_V_ channels, such as K_V_3 (KCNC) and K_V_2 (KCNB) channels; additionally, they are widely distributed and functionally expressed in neuroendocrine or endocrine cells [20,21,22,23,24,25]. Growing evidence reveals that there is a casual relationship with respect to the activity of K_V_3 or K_V_2 channels and the magnitude of delayed-rectifier K^+^ currents (*I*_K(DR)_), and that changes in current magnitude are thought to be correlated with action potential firing and neurotransmitter release [26,27,28,29,30,31,32]. Moreover, the K_V_ channels of K_V_2.1-K_V_3.2 types have been increasingly demonstrated and recognized as the main determinants of *I*_K(DR)_ identified in pituitary cells (e.g., pituitary GH_3_ cells) and neuroblastoma cells (e.g., NG108-15 cells) [20,24,28,33,34]. The biophysical properties of *I*_K(DR)_ intrinsically in GH_3_ cells are unique in being manifested not only by a positively shifted voltage dependency but also by a rapid deactivation rate [24,31,33,34]. The modulators of K_V_ channels have been noticeably demonstrated to possess antimanic action [35], while fluoxetine, a compound structurally similar to VOR, was effective at blocking astroglial Kir4.1-encoded currents [36,37]. However, current understanding of whether and how VOR or other related compounds result in any modifications with regard to the magnitude and/or gating kinetics of these types of K^+^ currents (e.g., *I*_K(DR)_) has not yet been thoroughly explored, although earlier reports have demonstrated the effectiveness of tricyclic antidepressants (e.g., desipramine, nortriptyline, and protriptyline) in the perturbations on voltage-gated K^+^ currents [38,39,40,41,42].

In light of the foregoing considerations, the goal of the present study was to examine the effect of VOR and other related compounds on ionic currents residing in pituitary GH_3_ cells with the aid of various voltage-clamp protocols. Of interest, we provide substantial evidence to disclose that, besides the inhibition of serotonin (5-HT) reuptake or binding to serotonin receptors, the presence of VOR can inhibit *I*_K(DR)_ in GH_3_ cells in a time-, state-, and concentration-dependent fashion at micromolar concentrations. The inhibitory effects on *I*_K(DR)_ caused by VOR are assumed to be direct and not mediated by bindings to serotonin or dopamine receptors, and they might be responsible for its pharmacological actions occurring in vivo. The K_V_ channels can thus be an important target for the action of this drug or other structurally related compounds.

## 2. Materials and Methods

### 2.1. Drugs, Chemicals and Solutions

Vortioxetine (VOR, Brintellix^®^ [worldwide], Trintellix^®^ (in USA and Canada), Lu AA21004, 1-(2-((2,4-dimethylphenyl)thio)phenyl) piperazine, C_18_H_22_N_2_S, (https://pubchem.ncbi.nlm.nih.gov/compound/Vortioxetine, accessed on 15 February 2022) was supplied by MedChemExpress (Genechain, Kaohsiung, Taiwan), while 4-aminopyridine (4-AP), E-4031, fluoxetine, propranolol, risperidone, serotonin (5-HT), and tetrodotoxin were supplied by Sigma-Aldrich (Merck, Taipei, Taiwan). Cell culture media, fetal bovine or calf serum, horse serum, L-glutamine, and trypsin-EDTA were acquired from HyClone^TM^ (Thermo Fisher, Waltham, MA, USA; Genechain). Unless stated otherwise, other chemicals and reagents (e.g., CdCl_2_, EGTA, and HEPES) were commercially available and of analytical grade.

The ionic compositions of extracellular solution (i.e., HEPES-buffered normal Tyrode’s solution) used in this work were as follows (in mM): NaCl 136.5, KCl 5.4, CaCl_2_ 1.8, MgCl_2_ 0.53, glucose 5.5, and HEPES-NaOH buffer 5.5 (pH 7.4). For measurements of *I*_K(DR)_, we kept GH_3_ or Neuro-2a cells immersed in Ca^2+^-free Tyrode’s solution in order to preclude the contamination by the magnitude of Ca^2+^-activated K^+^ currents and voltage-gated Ca^2+^ currents in these cells [43]. To record K^+^ currents (i.e., *I*_K(DR)_ or *I*_K(erg)_), we filled up the recording electrode with an internal solution consisting of (in mM): K-aspartate 130, KCl 20, KH_2_PO_4_ 1, MgCl_2_ 1, Na_2_ATP 3, Na_2_GTP 0.1, EGTA 0.1, and HEPES-KOH 5 (pH 7.2). To measure the current flowing through *I*_K(erg)_, cells were bathed in a high-K^+^, Ca^2+^-free solution (in mM): KCl 130, NaCl 10, MgCl_2_ 3, glucose 6, and HEPES-KOH 5 (pH 7.4). To measure voltage-gated Na^+^ or Ca^2+^ currents, we replaced K^+^ ions inside the pipette internal solution with equimolar Cs^+^ ions, and the pH was adjusted to 7.2 by adding CsOH. All solutions used in this work were prepared by using demineralized water from a Milli-Q purification system (Merck). The filling or bathing solution and culture media were filtered by using an Acrodisc^®^ syringe filter with Supor^®^ membrane (0.2 μm in pore size) (Genechain).

### 2.2. Cell Preparations

Both the pituitary adenomatous cell line, GH_3_ (BCRC-60015), and the mouse neuroblastoma cell line, Neuro-2a (BCRC-60026), were acquired from the Bioresource Collection and Research Center (Hsinchu, Taiwan). GH_3_ cells were grown in Ham’s F medium supplemented with 2.5% (*v*/*v*) fetal calf serum, 15% (*v*/*v*) horse serum, and 2 mM L-glutamine [34], while Neuro-2a cells were grown in Dulbecco’s modified Eagle’s medium with 10% (*v*/*v*) fetal bovine serum [44]. GH_3_ and Neuro-2a cells were maintained in a humidified atmosphere of 5% CO_2_ and 95% air at 37 °C. The subcultures were made in trypsinization (0.025% trypsin solution [HyClone^TM^] containing 0.01% sodium *N*,*N*-diethyldithiocarbamate and EDTA). The measurements were undertaken five or six days after cells were cultured up to 60–80% confluence.

### 2.3. Electrophysiological Measurements

During the few hours before the experiments, we dispersed GH_3_ or Neuro-2a cells by adding 1% trypsin-EDTA solution, and a few drops of cell suspension were quickly placed in a home-made recording chamber fixed on the stage of an inverted DM-II fluorescence microscope (Leica; Major Instruments, Tainan, Taiwan). Cells were immersed at room temperature (20–25 °C) in normal Tyrode’s solution, the composition of which was described above, and they were then allowed to attach to the chamber’s bottom before the measurements were made. The electrodes that we used were fabricated from Kimax-51 glass tubing with 1.5 mm outer diameter (#34500; Kimble, Dogger, New Taipei City, Taiwan) by using a vertical two-stage puller (PP-83; Narishige, Taiwan Instrument, Tainan, Taiwan). Filled with different internal solutions as described above, the electrodes used for measurements commonly had a tip resistance of 3–5 MΩ. During the recordings, they were firmly mounted in an air-tight holder with a suction port on the side, chloride silver wire was used to be in contact with the internal solution filled, and the electrode was delicately moved using a WR-6 hydraulic micromanipulator (Narishige). We examined ionic currents in the whole-cell configuration of a modified patch-clamp technique with the use of either an Axoclamp-2B (Molecular Devices, Sunnyvale, CA, USA) or an RK-400 amplifier (Bio-Logic, Claix, France), as described elsewhere [45,46,47]. GΩ-seals were achieved in an all-or-nothing fashion and resulted in a dramatic improvement in signal-to-noise-ratio. The liquid junction potentials, which occur when the ionic compositions in the pipette internal solution and those of bath solution are different, were zeroed shortly before GΩ-seal formation was made, and the whole-cell data were then corrected.

### 2.4. Data Recordings

The signals were monitored at a given interval and they were digitally captured and stored on-line at 10-kHz in an ASUS ExpertBook laptop computer (P2451F; Yuan-Dai, Tainan, Taiwan). For efficient analog-to-digital (A/D) and digital-to-analog (D/A) conversion to proceed, a Digidata^®^-1440A digitizer equipped with the laptop computer via a USB 2.0 interface was operated by pClamp 10.6 software run under Microsoft Windows 7 (Redmond, WA, USA). This device was connected to the computer with no need for a peripheral PC card. Current signals acquired were low-pass filtered at 2 kHz with a FL-4 four-pole Bessel filter (Dagan, Minneapolis, MN, USA). The pClamp-generated voltage-clamp protocols with various rectangular or ramp waveforms were designed and then delivered to the tested cells through D/A conversion in an effort to determine the current-voltage (*I-V*) relation or the steady-steady inactivation curve of ionic currents specified and the recovery time course of current inactivation. As pulse-train stimulation was applied to the tested cell, we used an Astro-Med Grass S88X dual output pulse stimulator (Grass, West Warwick, RI, USA). We placed the laptop computer on the top of an adjustable Cookskin stand (Ningbo, Zhejiang, China) laid out for ensuring the convenient operation during measurements.

### 2.5. Analyses of I_K(DR)_ Evoked by Membrane Depolarization

To assess the concentration-dependent effect of VOR on the peak and sustained amplitude of *I*_K(DR)_ in GH_3_ cells, we kept cells immersed in Ca^2+^-free Tyrode’s solution which contained 1 μM tetrodotoxin (TTX) and 0.5 mM CdCl_2_. Each tested cell was voltage-clamped at −50 mV and a 1-s depolarizing command voltage from −50 to +50 mV was applied to it. Current amplitudes at the beginning and end-pulse of the depolarizing pulse from −50 to +50 mV were measured in the control period (i.e., absence of VOR) and during cell exposure to different concentrations (0.3–100 μM) of VOR. The VOR concentration required to suppress 50% of the peak or sustained component of *I*_K(DR)_ activated by 1-s step depolarization from −50 to +50 mV was determined using a modified Hill function:$$\mathrm{Relative}\;\mathrm{amplitude}=\frac{{\left[VOR\right]}^{-n_{H}}\times \left(1-a\right)}{IC_{50}^{-n_{H}}+{\left[VOR\right]}^{-n_{H}}}+a$$
where, [*VOR*] is the *VOR* concentration used; n_H_ and *IC*_50_ are the Hill coefficient and the concentration required for 50% inhibition of peak or sustained *I*_K(DR)_, respectively. Maximal inhibition (i.e., 1 − *a*) was also evaluated. This equation can converge reliably to produce the best fit line and parameter estimates with an iterative least-squares method.

For the determination of the quasi-steady inactivation curve of *I*_K(DR)_ in the absence and presence of 10 or 30 μM VOR, we applied a two-step voltage-clamp protocol to the tested cells. In this set of experiments, a 1-s conditioning potential to a series of voltages ranging between −70 and +30 mV in 10-mV steps was applied to precede the 1-s test command voltage to +50 mV from a holding potential of −50 mV. The interval between two sets of voltage pulses was about 60 s to ensure complete recovery of *I*_K(DR)_. The relationships between the conditioning potentials and the normalized amplitudes of *I*_K(DR)_ in the absence and presence of 10 or 30 μM VOR were constructed by fitting the data to a modified Boltzmann function (or the Fermi–Dirac distribution). That is,
$$I=\frac{I_{max}}{1+\exp\left[\frac{\left(V-V_{1/2}\right)qF}{RT}\right]}$$
where *I*_max_ is the maximal amplitude of *I*_K(DR)_ in the absence and presence of 10 or 30 μM VOR; *V* and *V*_1/2_ represent the membrane potential in mV and the half-point in the inactivation curve of the current, respectively; *q* is the apparent gating charge in *e*; *F* Faraday’s constant; *R* the universal gas constant; and *T* the absolute temperature.

### 2.6. Statistical Analyses

Linear (first-order reaction binding) or nonlinear (exponential, Boltzmann or Hill function) curve fitting to experimental data sets was made with the iterative least-squares procedure by using different maneuvers, such as the Excel^®^-embedded “Solver” (Microsoft) and OriginPro^®^ 2021 program (OriginLab; Scientific Formosa, Kaohsiung, Taiwan). The averaged results are presented as the mean ± standard error of mean (SEM), with the size of experimental observations (*n*) denoting the cell numbers from which the data were collected. Paired or unpaired Student’s *t*-tests were initially used for the statistical analyses. When the statistical difference among varying groups was tested, we performed analysis of variance (ANOVA-1 or ANOVA-2) with or without repeated measures, followed by post-hoc Fisher’s least-significant difference test. Statistical analyses were performed by using the SPSS^®^ 20.0 package (AsiaAnalytics, Taipei, Taiwan). Statistical significance was determined at a *p* value of < 0.05.

## 3. Results

### 3.1. Effect of VOR on Delayed-Rectifier K^+^ Current (I_K(DR)_) Recorded from Pituitary GH_3_ Cells

For the first stage of experiments, we tested whether the presence of VOR exerts any perturbations on *I*_K(DR)_ evoked in response to the depolarizing command voltage from −50 to +50 mV with a duration of 1 s. Cells were placed in Ca^2+^-free Tyrode’s solution which contained 1 μM tetrodotoxin (TTX) and 0.5 μM CdCl_2_. The purpose of using this solution was to avoid interference by other types of ionic currents such as Ca^2+^-activated K^+^ current and voltage-gated Na^+^ or Ca^2+^ currents. As can be seen in Figure 1A, in the presence of VOR at a concentration of 1, 3 and 10 μM, upon 1-s membrane depolarization to +50 mV from a holding potential of −50 mV, the *I*_K(DR)_ was progressively decreased. For example, one minute after exposing GH_3_ cells to 3 μM VOR, the *I*_K(DR)_ amplitude measured at the end of depolarizing pulse was decreased to 512 ± 27 pA (n = 7, *p* < 0.05) from a control value of 648 ± 33 pA (n = 7). After VOR was removed, current amplitude was returned to 641 ± 31 pA (n = 7). In concurrence with these observations, the extent of VOR-mediated inhibition of *I*_K(DR)_ measured at the end of depolarizing test pulse (i.e., sustained component) was greater than that at the start of the pulse (i.e., peak component). For example, the presence of 3 μM VOR decreased the peak component of *I*_K(DR)_ from 823 ± 41 pA to 701 ± 36 pA (n = 7, *p* < 0.05). Moreover, in the continued presence of VOR (3 μM), a subsequent addition of E-4031 (10 μM), an inhibitor of *erg*-mediated K^+^ current (*I*_K(erg)_) [24], did not inhibit current amplitude further (698 ± 37 pA [in the presence of VOR] versus 697 ± 38 pA [in the presence of VOR plus E-4031]; n = 7, *p* > 0.05). As cells were continually exposed to 4-aminopyridine (4-AP, 5 mM), VOR-mediated modifications on *I*_K(DR)_ in GH_3_ cells remained effective. For example, in continued presence of 5 mM 4-AP, the *I*_K(DR)_ at the end of depolarizing pulse was significantly decreased from 644 ± 31 to 508 ± 25 pA (n = 7, *p* < 0.05) during the exposure to 3 μM VOR. The 4-AP was reported to be an inhibitor of A-type K^+^ currents [33].

We further constructed the concentration-dependent inhibition of peak and sustained *I*_K(DR)_ produced by the presence of VOR. Figure 1B depicts a differential concentration-dependent response of VOR-mediated reduction in peak and sustained components of *I*_K(DR)_. According to the modified Hill equation described in Materials and Methods, the IC_50_ values needed to exert an inhibitory effect on peak and sustained amplitude of the current were collated and then estimated to be 31.2 and 8.2 μM, respectively, indicating that these two values should be distinguishable.

It also needs to be mentioned that the suppressive effect of VOR on *I*_K(DR)_ in GH_3_ cells did not appear to emerge instantaneously (i.e., it varied in time), indicating that it occurred in a time- and concentration-dependent manner. In this regard, the VOR existence tends to fasten the inactivation time course of *I*_K(DR)_ evoked by 1-s maintained depolarization from −50 to +50 mV (Figure 1A). The current trajectories taken in different VOR concentrations were well fitted by single exponential with an iterative least-squares method. The presence of VOR at a concentration of 1, 3 and 10 μM was noticed to reduce the inactivation time constant (τ_inact_) of the current to 0.67 ± 0.04, 0.55 ± 0.03, and 0.33 ± 0.02 s (n = 7, *p* < 0.05), respectively, from a control value of 0.78 ± 0.05 s (n = 7).

In order to provide a quantitative estimate of VOR-induced block of *I*_K(DR)_ seen in GH_3_ cells, we next analyzed the time constants for its block on *I*_K(DR)_ during exposure to different VOR concentrations. The concentration dependence of the 1/∆τ value for *I*_K(DR)_ in response to sustained depolarization is illustrated in Figure 1C. It became clear from these results that cell exposure to VOR led to a concentration-dependent increase in the rate (1/∆τ) of current block. The forward (blocking) and backward (unblocking) rate constants derived from minimum binding scheme as described in Materials and Methods were consequently calculated to yield 0.168 s^−1^μM^−1^ and 1.32 s^−1^, respectively; therefore, the value of dissociation constant (K_D_ = k_−1_/k_+1_*) turned out to be 7.9 μM. This value is noticeably similar to the IC_50_ value required for VOR-mediated inhibition of sustained *I*_K(DR)_ described above; however, it is lower than that for its suppression of peak one.

### 3.2. Comparison in Effects on I_K(DR)_ Caused by Serotonin, VOR, VOR plus Propranolol, VOR plus Risperidone, and Fluoxetine

VOR has been previously demonstrated to have an agonistic or antagonistic effect on HT receptors [1,4,14,48], and it can also produce an increased release of serotonin, dopamine, and norepinephrine [2,49,50]. For these reasons, we attempted to determine whether VOR-mediated reduction of *I*_K(DR)_ can be influenced by these modulatory actions. As summarized in Figure 2, serotonin (10 μM) alone had little or no effect of *I*_K(DR)_ amplitude, while fluoxetine (10 μM), a structurally-similar antidepressant drug known to block K^+^ currents [36,37], mildly suppressed it. Moreover, in continued presence of 10 μM VOR, neither further application of propranolol (10 μM) nor risperidone (10 μM) can perturb VOR-induced reduction of *I*_K(DR)_ amplitude in these cells. Propranolol is an inhibitor of β-adrenergic and 5-HT receptors [51], and risperidone can suppress dopamine receptors [52]. Under our experimental conditions, these results reflect that, like the effect of fluoxetine, the VOR effect on *I*_K(DR)_ appears to be direct and unlinked to the inhibition of serotonin reuptake or the level of extracellular serotonin or dopamine.

### 3.3. Mean Current-Voltage (I-V) Relationship of Peak and Sustained Components of I_K(DR)_ Caused by VOR

In the next set of experiments, the *I*_K(DR)_’s evoked in response to a series of voltage pulses were examined to test whether the presence of VOR exerts any effects on the peak and sustain *I*_K(DR)_. In these experiments, when the whole-cell configuration was securely established, the voltage pulses ranging between −60 and +50 mV in 10-mV steps with a duration of 1 s were applied to the examined cells. As demonstrated in Figure 3A–C, cell exposure to VOR at a concentration of 10 μM resulted in a progressive reduction in *I*_K(DR)_ amplitude, which was concomitantly accompanied by an increase in inactivation time course of the current (i.e., a decrease in τ_inact_ value). The mean *I-V* relationships of peak or sustained I_K(DR)_ taken in the control period (i.e., absence of VOR) and during cell exposure to 10 μM are illustrated in Figure 3B,C, respectively. For example, when the depolarizing command voltage from −50 to +50 mV was applied, the presence of 10 μM VOR decreased the peak and sustained components of *I*_K(DR)_ to 412 ± 39 pA (n = 8, *p* < 0.05) and 161 ± 25 pA (n = 8, *p* < 0.05) from control values of 634 ± 51 pA (n = 8) and 489 ± 41 pA (n = 8), respectively. After the compound was removed, peak and sustained *I*_K(DR)_ returned to 631 ± 48 pA (n = 7) and 481 ± 38 pA (n = 7), respectively. Moreover, in the presence of 10 μM VOR, the peak and sustained whole-cell conductance of *I*_K(DR)_ measured at the voltage between +10 and +50 mV declined significantly to 6.71 ± 0.05 nS and 1.86 ± 0.03 nS (n = 8, *p* < 0.05) from the control values of 8.19 ± 0.06 nS and 6.48 ± 0.05 nS (n = 7), respectively. The observations, therefore, enable us to reflect that VOR exerts a depressant action on the peak and sustained components of *I*_K(DR)_ intrinsically in GH_3_ cells, and that this drug tends to be selective for sustained over peak *I*_K(DR)_ evoked in response to depolarizing command voltages.

### 3.4. Steady-State Inactivation Curve of I_K(DR)_ Produced by the Presence of VOR

In order to characterize the inhibitory effect of VOR on *I*_K(DR)_, we further tested whether the exposure to different VOR concentrations might lead to changes in the inactivation curve of *I*_K(DR)_ identified in GH_3_ cells. Figure 4 illustrates the inactivation curve of *I*_K(DR)_ in the absence of VOR and during exposure to 10 or 30 μM VOR. The sigmoidal curve derived from the data sets was appropriately fitted with the modified Boltzmann equation (stated under Materials and Methods). That is, the value of *V*_1/2_ taken in the control period was −41 ± 3 mV (n = 8), while those in the presence of 10 and 30 μM VOR were −38 ± 3 mV (n = 8) and -53 ± 3 mV (n = 8), respectively; however, the *q* value in the control period was 5.2 ± 0.7 *e* (n = 8), and those during exposure to 10 and 30 μM VOR were 5.3 ± 0.7 *e* (n = 8) and 5.1 ± 0.7 *e* (n = 8), respectively. The results enable us to disclose that the presence of 30 μM VOR could cause a leftward shift along the voltage axis in the inactivation curve by approximately 15 mV, as compared with. the effect caused by 10 μM VOR on the curve. However, the gating charge of the current in the presence of 10 and 30 μM VOR did not differ significantly.

### 3.5. Change in Current Inactivation Time Course Produced by VOR

The presence of VOR not only decreased the amplitude of peak and sustained *I*_K(DR)_, but it also accelerated the inactivation time course of the current during maintained depolarization. As such, we further explored how the current decay during exposure to VOB can be altered by different level of test pulses with varying durations. In this set of voltage-clamp experiments, the rectangular depolarizing pulse from −50 to +50 mV with varying durations which was then followed by a return to the level of 0 mV for 300 ms was applied to evoke *I*_K(DR)_ in the VOR presence. Figure 5 illustrates a representative record revealing that, during cell exposure to 10 μM VOR, the inactivation trajectory of *I*_K(DR)_ evoked by varying durations of test pulse was decayed in an exponential fashion. One minute after exposing cells to 10 μM VOR, the fast and slow time constants of current decline obtained at the level of +50 mV were estimated to yield 37.9 ± 1.1 and 211 ± 18 ms (n = 7), respectively. However, at the level of 0 mV (i.e., return to 0 mV), the same concentration of VOR caused the decaying time constant to become a single exponential process with the time constant of 92.5 ± 2.4 ms (n = 7). The experimental observations thus reflect that the VOR existence can alter current inactivation of *I*_K(DR)_ evoked by varying durations of depolarizing pulse in a one- or two-exponential fashion.

### 3.6. I_K(DR)_ Recovery from Inactivation in the Presence of VOR

Recovery from block was further determined by using a two-step protocol consisting of a first (conditioning) depolarizing pulse of adequate length to allow block to reach a steady state. In the control period or during cell exposure to VOR (10 or 30 μM), the membrane potential was thereafter stepped to +50 mV from −50 mV for a variable time with a geometric progression, after which a second depolarizing command voltage (test pulse) was applied at the same potential as the conditioning pulse (Figure 6A). The normalized amplitudes (i.e., the peak components of *I*_K(DR)_ evoked in response to the test pulse were divided by those by the conditioning pulse) were taken as a measure of recovery from current inactivation, and they were constructed and plotted versus interpulse interval (Figure 6B). In the control period, the *I*_K(DR)_ recovery from inactivation was estimated by a single exponential with a time constant of 0.050 ± 0.04 s (n = 7). As cells were exposed to 10 or 30 μM VOR, recovery of current inactivation was noticeably complete when the interpulse interval was around 10 s, and in the presence 10 and 30 μM VOR, the time constants of current recovery from inactivation were estimated to be 0.51 ± 0.04 s (n = 7) and 1.34 ± 0.06 s (n = 7), respectively. It is clear from the present observations that increasing VOR concentration can slow down the recovery from current inactivation identified in GH_3_ cells.

### 3.7. Effect of VOR on I_K(DR)_ Activated by Varying Downsloping Ramp Pulse

It is known that the magnitude of *I*_K(DR)_ contributes significantly to the repolarizing phase of action potentials in excitable cells [29,32]. We also decided to determine how the *I*_K(DR)_ activated by the descending ramp pulse with varying duration can be altered in the presence of VOR. As demonstrated in Figure 7, upon the downsloping ramp pulse, the instantaneous amplitude of *I*_K(DR)_ measured at the voltage between −10 and +50 mV with the ramp duration of 0.4 or 1.6 s appears to be different. For example, during cell exposure to 10 μM VOR, current amplitude measured at the level of +30 mV with a ramp duration of 0.4 s was 297 ± 23 pA (n = 7), while that at the same level with a ramp duration of 1.6 s was significantly smaller (165 ± 19 pA; n = 7, *p* < 0.05). Furthermore, as the VOR concentration was increased to 30 μM, the current magnitude induced by the downsloping ramp pulse was further decreased; however, no change of the difference in current magnitude measured between the ramp duration of 0.4 and 1.6 s was demonstrated. It is reasonable to assume, therefore, that the *I*_K(DR)_ magnitude inhibited by adding VOR is subject to be modified during the varying falling phase of action potential.

### 3.8. VOR-Induced Increase in Cumulative Inhibition of I_K(DR)_ Inactivation

*I*_K(DR)_ inactivation was previously demonstrated to accumulate activated during repetitive short pulses [26,27]. Further measurements were taken to explore whether the presence of VOR modifies the inactivation process of the current evoked during repetitive stimulation. The examined cell was voltage-clamped at −50 mV, and a stimulus protocol consisting of repetitive depolarization to +50 mV (100 ms in each pulse with a rate of 5 Hz for 10 s) was applied to it. As revealed in Figure 8, when cells were exposed to 10 μM VOR, the current inactivation was evoked by a 10-s sustained depolarization from −50 to +50 mV with the inactivation time constant of 2.21 ± 0.12 and 0.14 ± 0.04 s (n = 7) (i.e., decay with a two-exponential process). During exposure to the same concentration, in addition to a reduction in current magnitude, the exponential time course of *I*_K(DR)_ evoked by a train of depolarizing pulse became shortened to 1.94 ± 0.11 and 0.11 ± 0.03 s (n = 7). Furthermore, the presence of 30 μM VOR was found to decrease the decaying time constant further to 1.72 ± 0.11 and 0.09 ± 0.03 s (n = 7). Therefore, the results enable us to reflect that, apart from its decrease in current magnitude, during cell exposure to VOR, the decay of *I*_K(DR)_ evoked by a 10-s train of depolarizing pulses (i.e., accumulative inactivation of the current) can be enhanced.

### 3.9. Effect of VOR on Erg-Mediated K^+^ Current (I_K(erg)_) in GH_3_ Cells

VOR has been previously reported to exert a prolongation in the QTc interval [53,54]. Desipramine, known to be an antidepressant drug, was demonstrated to suppress hERG K^+^ channels [39,40]. The magnitude of *I*_K(erg)_ is strongly linked to the cardiac repolarization and to the lengthening in QTc interval. For these reasons, we further tested whether VOR could lead to any modifications on *I*_K(erg)_, which has been demonstrated to be functionally expressed in GH_3_ cells [20,24,55,56]. In these experiments, we bathed cells in high-K^+^, Ca^2+^-free solution and the electrode was filled with K^+^-containing solution. The tested cells were held at −10 mV and a series of 1-s voltage pulses from −90 to 0 mV in 10-mV increments were applied to evoke the deactivating *I*_K(erg)_. As demonstrated in Figure 9A,B, the *I*_K(erg)_ amplitudes measured at the start of each voltage pulse throughout the entire voltage-clamp steps were evidently lessened by adding VOR at a concentration of 10 μM. For example, one minute after GH_3_ cells were exposed to 10 μM VOR, the peak amplitude of deactivating *I*_K(erg)_ at the level of −90 mV was decreased to 294 ± 21 pA (n = 8, *p* < 0.05) from a control value of 399 ± 28 pA (n = 8). After a washout of VOR, current amplitude returned to 391 ± 26 pA (n = 8). Additionally, upon cell exposure to 10 μM VOR, the whole cell conductance of *I*_K(erg)_ measured in the voltage between −90 and −50 mV was measurably decreased to 3.58 ± 0.04 nS (n = 8, *p* < 0.05) from a control value of 4.06 ± 0.03 nS (n = 8). Therefore, as with the *I*_K(DR)_ described above, the *I*_K(erg)_ identified from GH_3_ cells is vulnerable to inhibition by VOR.

### 3.10. Inhibitory Effect of VOR on I_K(DR)_ Residing in Neuro-2a Neuroblastoma Cells

VOR has been tailored for the management of depressive disorders [5,6,7,8,9,10,11]. Therefore, in a final set of experiments, we experimented on a different type of electrically excitable cells (i.e., Neuro-2a cells). The preparation of Neuro-2a cells was described in Materials and Methods. After being allowed to attach to the chamber’s bottom, cells were bathed in Ca^2+^-free Tyrode’s solution and the recording electrode was filled with K^+^-enriched solution. As demonstrated in Figure 10, one minute after Neuro-2a cells were exposed to VOR (3 or 10 μM), the amplitude of *I*_K(DR)_ was progressively decreased, which was accompanied by an increase in the inactivation time course of the current. Additionally, in the continued presence of 10 μM VOR, subsequent addition of propranolol (10 μM) failed to modify VOR-mediated inhibition of *I*_K(DR)_. Propranolol was previously shown to block β-adrenergic and 5-HT receptors [51]. Furthermore, as cells were continually exposed to 4-AP (5 mM) or E-4031 (10 mM), VOR-mediated modification of *I*_K(DR)_ amplitude seen in these cells remained effective. Additionally, in continued presence of 10 mM VOR, further addition of 5 mM 4-AP or E-4031 (10 mM) did not alter *I*_K(DR)_ amplitude further in Neuro-2a cells (285 ± 17 pA [in the presence of VOR], 284 ± 18 pA [in the presence of VOR plus 4-AP], versus 286 ± 19 pA [in the presence of VOR plus E-4031]; n = 7, *p* > 0.05). Therefore, as with VOR actions on *I*_K(DR)_ in GH_3_ cells, cell exposure to VOR is effective at suppressing *I*_K(DR)_ in Neuro-2a cells.

## 4. Discussion

The noticeable findings from the present study are that (a) the presence of VOR could depress the *I*_K(DR)_ amplitude in a concentration-, time-, and state-dependent manner identified in GH_3_ cells; (b) cell exposure to this compound led to the differential inhibition of peak or sustained component of *I*_K(DR)_ activated by long depolarizing pulse with effective IC_50_ values of 31.2 and 8.5 μM, respectively; (c) the peak or sustained *I*_K(DR)_ measured throughout the entire voltage-clamp range was depressed during exposure to VOR; (d) increasing VOR concentration produced a leftward shift in the steady-state inactivation curve of *I*_K(DR)_; (e) the VOR presence produced current inactivation at +50 mV or +0 mV with two- or one-exponential process, respectively; (f) the *I*_K(DR)_ evoked by the downsloping ramp pulse can be perturbed during exposure to VOR; (g) the accumulative inactivation of *I*_K(DR)_ evoked by repetitive stimuli was enhanced by VOR; (h) cell exposure to VOR decreased the amplitude of *I*_K(erg)_; and (i) in mouse Neuro-2a cells, VOR suppressed *I*_K(DR)_ in combination with an increase of current inactivation rate. However, GH_3_-cell exposure to 10 μM VOR caused a mild inhibition in voltage-gated Na^+^ or L-type Ca^2+^ currents by about 5%. Together with these data, the modulation of transmembrane ionic currents exerted by VOR conceivably represents a part of the fundamentally molecular mechanisms through which it possesses pharmacological properties and functional influence in various neurological disorders (e.g., major depression and other neurodegenerative disorders) [35].

The values of K_D_ (calculated from the first-order reaction scheme) and IC_50_ (required for VOR-mediated inhibition of sustained *I*_K(DR)_) obtained from GH_3_ cells were virtually indistinguishable (i.e., around 8 μM). Previous studies have shown that VOR had a functional antagonist at the r5-HT_7_ receptor with an IC_50_ of 2.1 μM [49], although this drug showed high-affinity binding for the h5-HT_3A_ receptor and potent functional antagonism at cloned rat and human 5-HT_3A_ receptors [1]. Fluoxetine, a compound with structural similarity to VOR, was also found to suppress K^+^ currents effectively [36,37]. It is, therefore, plausible to reflect that the concentration required for the inhibition of *I*_K(DR)_ in different excitable cells overlaps with that for its multimodal effects on serotonin (5-HT) receptors [49]. Alternatively, given that the VOR molecules can penetrate the blood–brain barriers, the unanticipated modifications of *I*_K(DR)_ and *I*_K(erg)_ demonstrated herein are expected to occur at a concentration achievable in humans.

In the present study, the addition of serotonin (5-HT) to the bath could not modify the *I*_K(DR)_ amplitude evoked by membrane depolarization in GH_3_ cells, while that of fluoxetine mildly suppressed it. Moreover, as cells were continually exposed to VOR, neither subsequent application of propranolol nor that of risperidone was able to modify VOR-mediated inhibition of *I*_K(DR)_. Propranolol can exert antagonistic effects on both β-adrenergic and 5-HT receptors [51], while risperidone can block dopamine receptors [52]. The experimental results thus prompted us to reflect that the inhibition of *I*_K(DR)_ and *I*_K(erg)_ exerted by the existence of VOR could be direct and independent of its actions on either binding to 5-HT receptors or suppression on serotonin reuptake [1,2,3,4,14,35,48,49,50].

It needs to be mentioned that the increasing VOR concentration not only produced a leftward shift in the steady-state inactivation curve of *I*_K(DR)_ but also slowed the recovery of *I*_K(DR)_ block observed in GH_3_ cells. The accumulative inhibition of *I*_K(DR)_ activated during a train of depolarizing pulses was also enhanced during cell exposure to VOR. As such, findings from the present results allow us to imply that the increase of *I*_K(DR)_ inactivation exerted by its presence has the propensity to increase a process of long-lasting facilitation or potentiation during the firing of action potentials in excitable cells [23,28,30,31]. In Neuro-2a cells, the exposure to VOR resulted in the suppression of *I*_K(DR)_ together with a measurable shortening in the τ_inact_ value of the current. Therefore, whatever the detailed ionic mechanism of its actions, the experimental observations in this study strongly suggest that the inhibitory effect of VOR on *I*_K(DR)_ is primarily mediated through a state-dependent, open-channel block mechanism. Block by VOR of *I*_K(DR)_ presented herein is of peculiar importance, since it may have characteristics rendering it potentially significant from either a pathophysiological, therapeutic, or toxicological point of view. However, whether or how VOR-mediated changes in *I*_K(DR)_ could occur in cultured primary neurons requires further study.

In a previous study, the maximal plasma concentration following multiple doses of 5, 10, or 20 mg was estimated to be 9 (0.024 μM), 18 (0.047 μM), and 33 ng/mL (0.087 μM) [57]. Other reports demonstrated that maximal plasma level following the VOR administration could reach 50 ng/mL (0.13 μM) [49,58]. Those values tend to be lower than the IC_50_ or K_D_ value for VOR-mediated inhibition of *I*_K(DR)_ described herein. Moreover, it needs to be emphasized that the action of VOR on excitable cells can depend on varying confounding factors, including the preexisting level of resting potential, different discharge patterns of action potentials (e.g., bursting pattern or high firing rate), the VOR concentration achieved, and in any combinations of these three [30,31,32,35]. It is therefore possible that both *I*_K(erg)_ and *I*_K(DR)_ inhibited by VOR can concertedly act to participate in its perturbations on the behaviors of excitable cells. The blood–brain barrier penetration was also confirmed by measuring target occupancy of VOR on the serotonin receptor after its administration [49]. In such a scenario, it is anticipated that the perturbations on ionic currents exerted by the VOR addition could be of pharmacological, therapeutic, or toxicological relevance [35,59].

Earlier reports have demonstrated the capability of VOR to prolong QTc interval on the electrocardiogram [53,54], thereby leading us to propose that VOR-mediated inhibition of K^+^ currents could possibly be linked to its action on the heart function (e.g., cardiac repolarization), as reported previously for desipramine-mediated suppression of hERG channels [39,40]. To what extent VOR-induced adverse effects (e.g., nausea, constipation, and vomiting) [14] are closely connected with its inhibition of K^+^ currents remains to be further evaluated. Noticeably, additional evidence in the in vivo conditions, such as the electrophysiological recordings and the serotonin concentration measurements in cerebrospinal fluid, still needs to be assessed to support the notion that VOR-induced changes in K^+^ currents are independent of its binding to serotonin receptors.

It needs to be stressed that the model GH_3_ cells used in this study might not be suitable for investigations on the antidepressant or anxiolytic effects of VOR, since the neuronal target of this drug is incompletely understood. Therefore, it still remains unclear which distinct types of ionic channels can be affected by VOR or whether such channels involved are expressed in the neuronal targets of the antidepressant VOR.

## Figures and Tables

**Figure 1 biomedicines-10-01318-f001:**
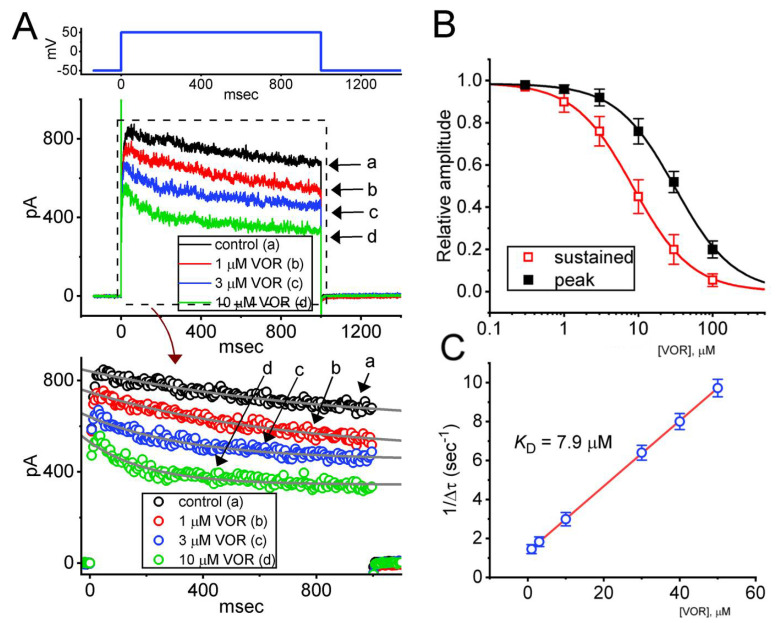
Inhibitory effect of VOR on delayed rectifier K^+^ current (*I*_K(DR)_) measured from pituitary GH_3_ cells. In this set of measurements, we kept cells bathed in Ca^2+^-free Tyrode’s solution, and the electrode that we used was filled up with a K^+^-containing solution. The ionic compositions in the solutions used were described in Materials and Methods. (**A**) Representative current traces obtained in the control period (a, i.e., absence of VOR) and during cell exposure to 1 μM VOR (b), 3 μM VOR (c), and 10 μM VOR (d). The voltage-clamp protocol applied is illustrated in the top part. The lower part indicates expanded record from current traces in the upper part (dashed box). Each continuous gray line denotes the best fit to current trajectory (open circle symbols) in the absence and presence of VOD to single exponential function. (**B**) Concentration-dependent inhibition of peak (filled black square symbols) and sustained (open red square symbols) components of *I*_K(DR)_ caused by VOR. The peak or sustained component of *I*_K(DR)_ was respectively measured at the start or the end-pulse of the depolarizing command voltage from −50 to +50 mV with a duration of 1 s. Each data point indicates the mean ± SEM (n = 8). The continuous sigmoidal curve was optimally created by fitting the data points to a modified Hill function by using an iterative least-squares procedure. The Hill equation was detailed in Materials and Methods. The effective IC_50_ values required for VOR-mediated inhibition of peak and sustained *I*_K(DR)_ were computed to be 31.2 and 8.5 μM, respectively. (**C**) The linear relationship of the 1/∆τ value versus the VOR concentration (mean ± SEM; n = 7 for each point). According to the first-order reaction scheme (elaborated in Materials and Methods), the forward (k_+1_*) and backward (k_−1_) rate constants for VOR-mediated interaction with *I*_K(DR)_ activated by step depolarization were calculated to be 0.168 s^−^^1^μM^−^^1^ and 1.32 s^−^^1^, respectively; meanwhile, the K_D_ (i.e., k_−__1_/k_+1_*) yielded was 7.9 μM. The continuous red line indicates best fit to a linear regression.

**Figure 2 biomedicines-10-01318-f002:**
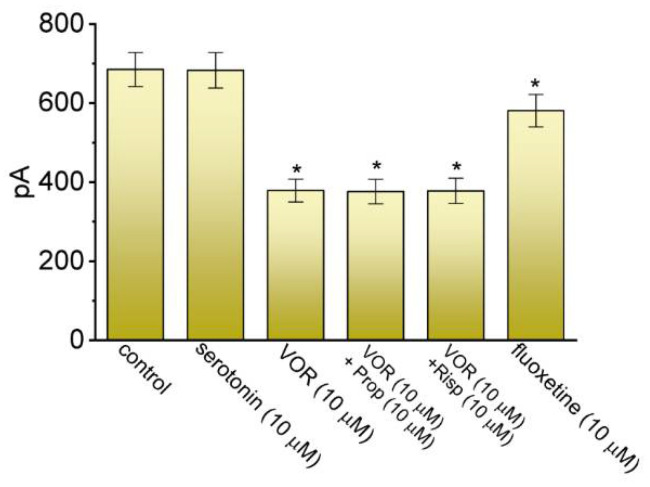
Comparison in inhibitory effects on *I*_K(DR)_ among serotonin, VOR, VOR plus propranolol (Prop), VOR plus risperidone (Risp) and fluoxetine in GH_3_ cells. Current amplitude measured at the end of 1-s depolarizing pulse from −50 to +50 mV. Each bar indicates the mean ± SEM (n = 7). * Significantly different from control (*p* < 0.05). Of note, the serotonin presence has no effect on *I*_K(DR)_ amplitude, while fluoxetine mildly suppresses it; furthermore, the addition of neither propranolol nor risperidone, still in the presence of VOR, does not reverse VOR-mediated inhibition of *I*_K(DR)_.

**Figure 3 biomedicines-10-01318-f003:**
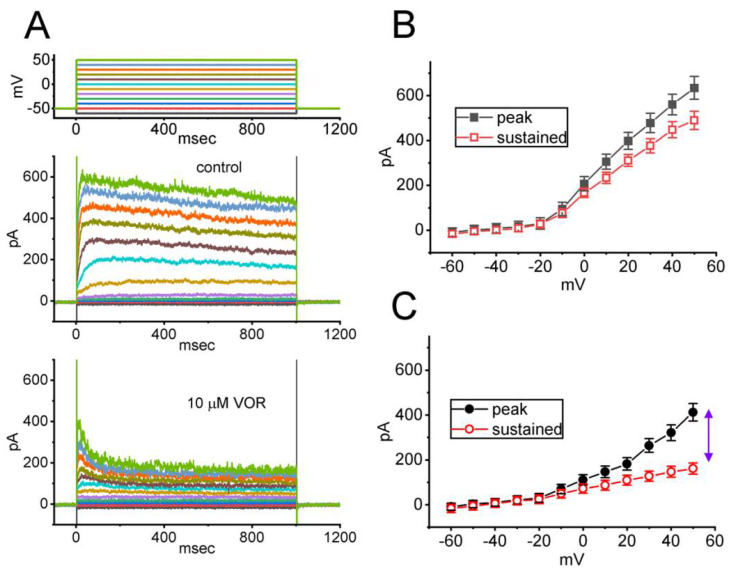
Mean current-voltage (*I-V*) relationship of the peak and sustained components of *I*_K(DR)_ acquired with or without the VOR addition. The experiments were conducted in cells immersed in Ca^2+^-free Tyrode’s solution, and we filled up the electrode with K^+^-containing solution. (**A**) Representative current traces acquired in the absence (upper) and presence (lower) of 10 μM VOR. Voltage-clamp protocol that we used is illustrated in the uppermost part. In (**B**,**C**), the mean *I-V* relationships of the peak (filled black symbols) or sustained (open red symbols) component of *I*_K(DR)_ were respectively taken in the control period (i.e., VOR was not present) and during exposure to 10 μM VOR. Each point represents the mean ± SEM (n = 8 for each point). The double arrowhead in the right side of (**C**) indicates the larger difference in current magnitude between peak and sustained components of *I*_K(DR)_ in the presence of 10 μM VOR.

**Figure 4 biomedicines-10-01318-f004:**
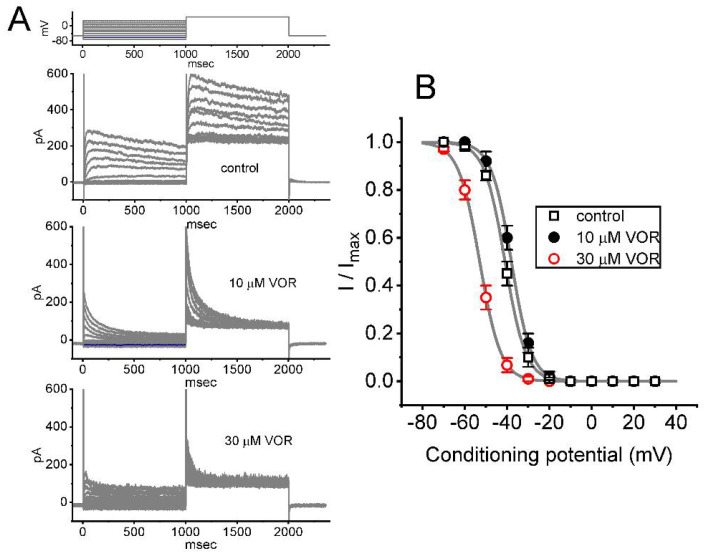
The steady-state inactivation curve of *I*_K(DR)_ in the control period (i.e., VOR was not present) and during cell exposure to 10 or 30 μM VOR. The experiments were conducted with a two-step voltage protocol (indicated in the uppermost part of (A)). (**A**) Representative current traces obtained in the absence (upper) and presence of 10 μM VOR (middle) or 30 μM VOR (lower). (**B**) The relationships of the conditioning potential versus the normalized amplitude of *I*_K(DR)_ in the control period (open black squares) and during cell exposure to 10 μM VOR (filled black circles) or 30 μM VOR (open red circles). The data represents the mean ± SEM (n = 8 for each point). The continuous curves were obtained with the goodness-of-fit test by a Boltzmann function, as described in Materials and Methods.

**Figure 5 biomedicines-10-01318-f005:**
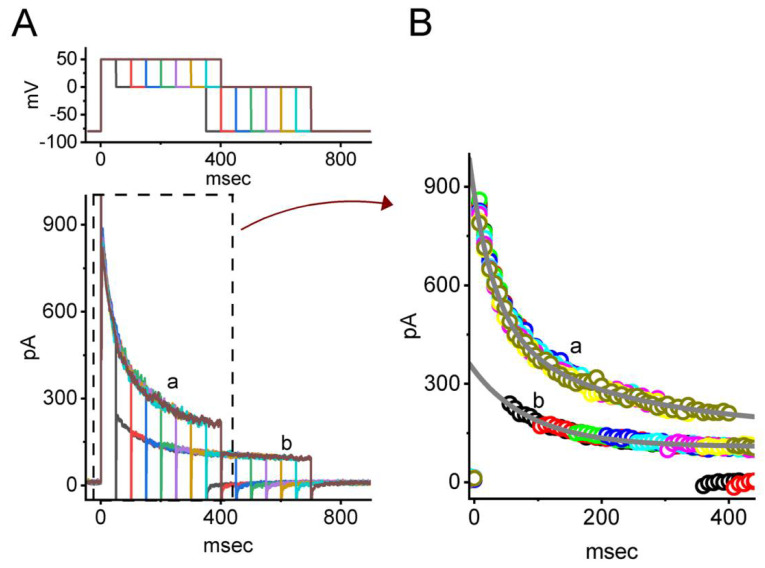
Effect of VOR on *I*_K(DR)_ evoked by the depolarizing pulse with varying durations. These experiments were conducted in cells held at the level of −50 mV, and the depolarizing command voltages from −50 to +50 mV with different durations of test pulse (50–400 ms in 50-ms increments) followed by a return to 0 mV for 300 ms were applied to the examined cell. In (**A**), current traces labeled a were obtained at the level of +50 mV with varying duration during cell exposure to 10 μM VOR, while the traces labeled b were at the level of 0 mV. The voltage-clamp protocol applied is illustrated in the upper part. In (**B**), current traces indicate an expanded record (dashed box) in (A). The time constants obtained at the level of +50 mV (a) by fitting to two exponential function was well estimated to be 211 and 37.9 ms, while that at 0 mV (b) by fitting to single exponential was 92.5 ms.

**Figure 6 biomedicines-10-01318-f006:**
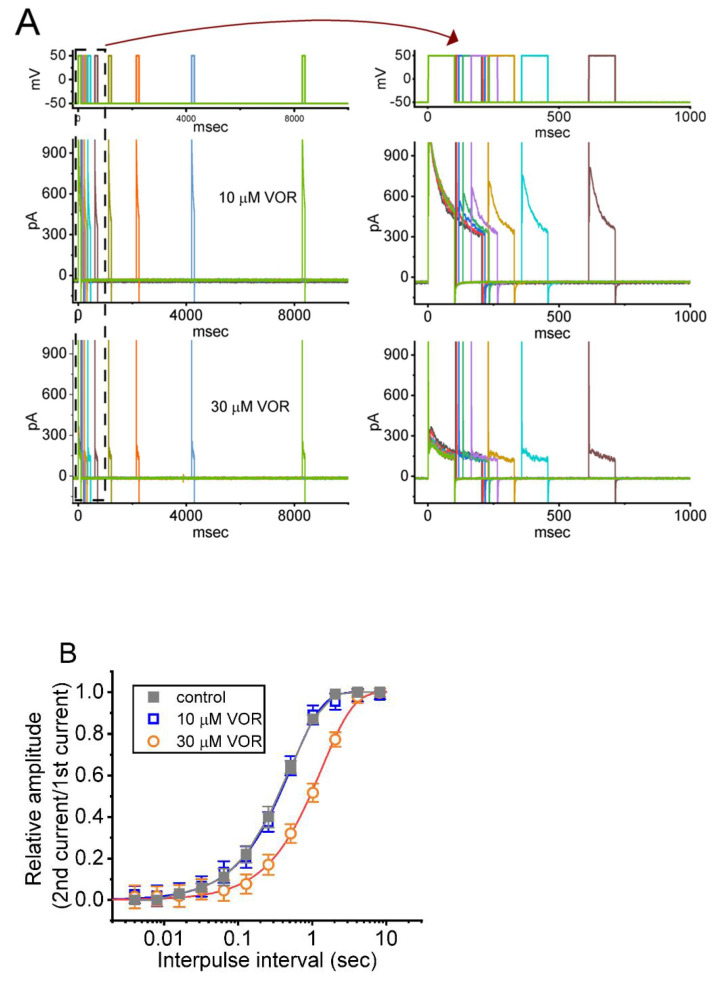
The time course of the recovery of *I*_K(DR)_ inactivation in the presence of 10 and 30 μM VOR. In these experiments, cells, bathed in Ca^2+^-free Tyrode’s solution, were depolarized from −50 to +50 mV with a duration of 100 ms, and varying interpulse durations with a geometric progression were applied to the tested cells. (**A**) Representative current traces obtained in the presence of 10 μM VOR (upper) and 30 μM VOR (lower). The uppermost parts denote the voltage-clamp protocol applied. The right side in (**A**) shows the expanded records (i.e., potential and current traces) taken from the dashed box in the left side for a short time scale. (**B**) The mean relationship of the relative amplitude versus the interpulse interval in the absence (filled gray squares) and presence of 10 mM VOR (open blue squares) or 30 μM VOR (open red circles). The x-axis is displayed on the logarithmic scale. The time course as indicated in smooth line was goodness-of-fit to a single exponential. Each point represents the mean ± SEM (n = 7).

**Figure 7 biomedicines-10-01318-f007:**
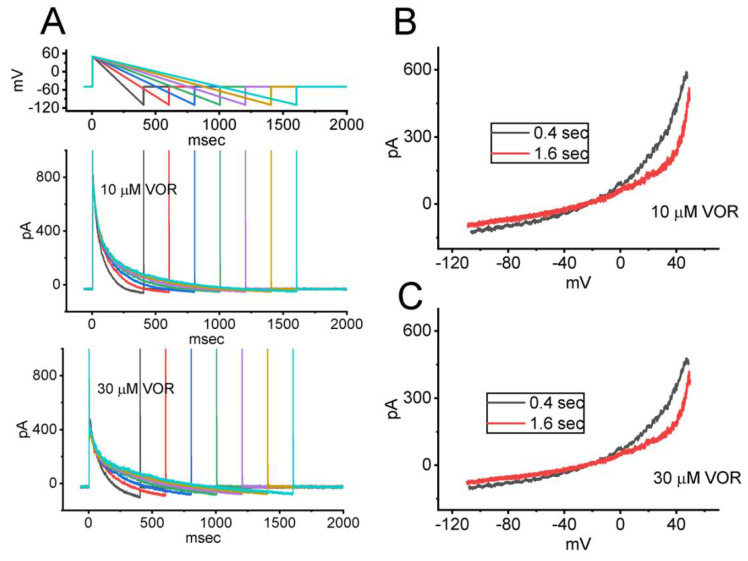
Effect of VOR on *I*_K(DR)_ evoked by the descending ramp pulse with varying durations in GH_3_ cells. In these experiments, the *I*_K(DR)_ taken in the presence of 10 or 30 μM VOR was evoked by the depolarizing pulse from −50 to +50 mV followed by the descending (repolarizing) voltage pulse from +50 to −110 mV with varying duration ranging between 0.4 and 1.6 s (i.e., ramp slope between −0.4 and −0.1 V/s). (**A**) Representative current traces acquired during cell exposure to 10 μM VOR (upper) or 30 μM VOR (lower). The uppermost part indicates the voltage-clamp protocol applied. In (**B**,**C**), the relationships of instantaneous *I*_K(DR)_ amplitude versus ramp voltage achieved with a ramp duration of 0.4 s (black color) or 1.6 s (red color) were respectively demonstrated in the presence of 10 and 30 μM VOR. Notably, as cells were exposed to VOR, during such downsloping ramp pulse, the current amplitude taken at the voltage range between −10 and +50 mV with the ramp duration of 0.4 and 1.6 s tends to be distinguishable; however, increasing VOR concentration from 10 to 30 μM significantly reduced *I*_K(DR)_ magnitude with no evident change in the difference of current amplitude between different ramp durations applied (i.e., 0.4 and 1.6 s).

**Figure 8 biomedicines-10-01318-f008:**
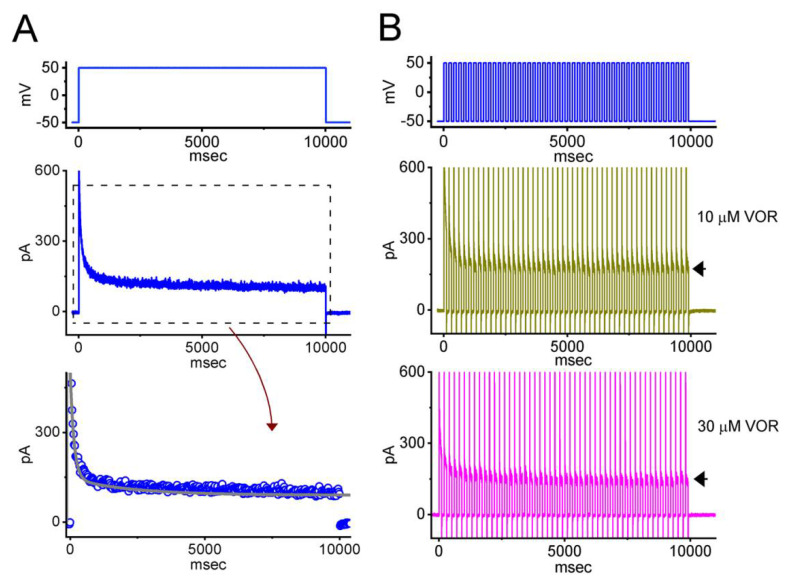
Excessive accumulative inactivation of *I*_K(DR)_ in the presence of VOR recorded from GH_3_ cells. In this set of experiments, currents were evoked during repetitive depolarization to +50 mV from a holding potential of −50 mV with a duration of 10 s. The depolarizing pulses used to evoke *I*_K(DR)_ lasted 100 ms at a rate of 5 Hz. In (**A**)**,** representative current trace evoked by 10-s maintained depolarization from −50 to +50 mV (indicated in the uppermost part) was obtained during cell exposure to 10 μM VOR. The lower part shows expanded time scale of current trace (dashed box in the upper part), and smooth gray curve was appropriately fitted to a two-exponential function in which the decaying time constants with 2.2 s (slow component) and 0.14 s (fast component) were obtained. In (**B**), representative current traces evoked by repetitive depolarization from −50 to +50 mV were acquired in the presence of 10 μM VOR (upper part) and 30 μM VOR (lower part), respectively. The uppermost part in (**B**) demonstrates the voltage-clamp protocol applied. Arrowheads in (**B**) denote the occurrence of *I*_K(DR)_ trajectories activated in response to repetitive stimuli. Notably, in addition to the inhibition of *I*_K(DR)_ amplitude, the VOR presence raises the rate of excessive accumulative inactivation of *I*_K(DR)_ evoked by repetitive stimuli.

**Figure 9 biomedicines-10-01318-f009:**
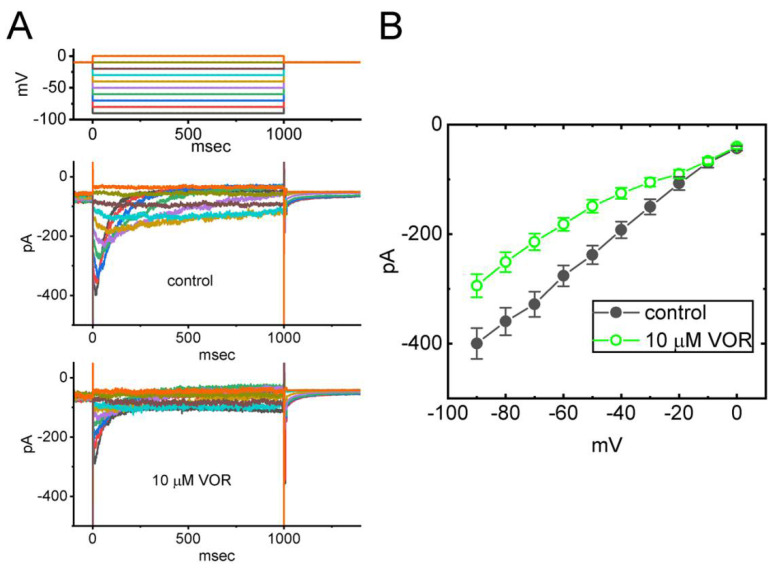
Effect of VOR on *erg*-mediated K^+^ current (*I*_K(erg)_) identified in GH_3_ cells. This stage of experiments was undertaken in cells bathed in high-K^+^, Ca^2+^-free solution, and the electrode was filled with K^+^-containing solution. The examined cell was voltage-clamped at −10 mV, and a series of voltage pulses between −90 and 0 mV was applied to it. (**A**) Representative current traces obtained in the control period (upper, i.e., absence of VOR) and during exposure to 10 μM VOR (lower). The uppermost part is the voltage protocol used. (**B**) Mean *I-V* relationship of deactivating *I*_K(erg)_ in the absence (filled black symbols) and presence (open green symbols) of 10 μM VOR (mean ± SEM; n = 8 for each point). The peak *I*_K(erg)_ amplitudes with or without the VOR addition were measured at the start of each voltage step.

**Figure 10 biomedicines-10-01318-f010:**
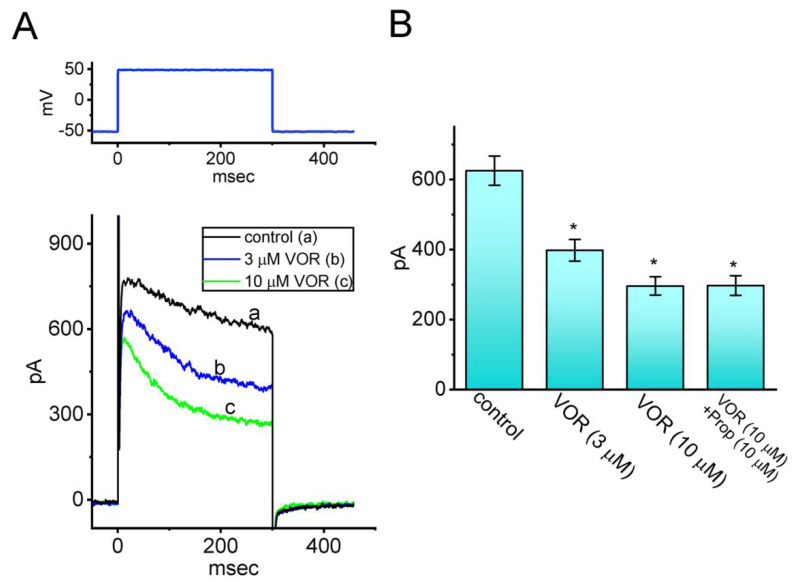
Inhibitory effect of VOR on *I*_K(DR)_ identified from Neuro-2a cells. In these experiments, cells were bathed in Ca^2+^-free Tyrode’s solution containing 1 μM TTX and 0.5 mM CdCl_2_. (**A**) Representative current trace obtained in the control period (a, i.e., absence of VOR) and during cell exposure to 3 μM VOR (b) or 10 μM VOR (c). The upper part shows the voltage protocol used. (**B**) Summary bar graph demonstrating effect of VOR and VOR plus propranolol (Prop) on the amplitude of *I*_K(DR)_ (mean ± SEM, n = 7 for each bar). Current amplitude was measured at the end of 300-ms depolarizing pulse from −50 to +50 mV. * Significantly different from control (*p* < 0.05).

## Data Availability

The original data is available upon reasonable request to the corresponding author.

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
