# Peer review of "Inhibitory Effectiveness in Delayed-Rectifier Potassium Current Caused by Vortioxetine, Known to Be a Novel Antidepressant"

_biomedicines, 2022, doi:10.3390/biomedicines10061318_

Round 1

Reviewer 1 Report

The authors conducted sophisticated experiments to describe the effects of the antidepressant drug Vortixetine (VOR) on ionic currents in rat pituitary GH3 cells. According to their results, they conclude that VOR suppressed delayed-rectifier K+ currents (IKdr) whereby VOR differentially inhibited the peak and sustained components of this current. Furthermore, they conclude that the modulation of currents by VOR conceivably represents a part of the mechanisms through which it possessed pharmacological properties and functional influence in various neurological disorders (e.g., major depression).

I have several major problems with the presented data and the conclusions drawn.

  1. The main issue is the type of current inhibited by VOR and the resulting effect on the current kinetics. GH3 cells express not only delayed rectifier but also inactivating K+ channels (IKA type). If VOR inhibits selectively IKdr, the remaining IKA current will change the kinetics of the activated currents and this is not caused by an influence of VOR on IKdr kinetics. Therefore, the authors must show that their currents are mediated by IKdr only, for example by application of 4-AP. The same applies to the experiments on Neuro-2a cells.

  1. The authors should show to which extend their VOR-blocked IKdr current is mediated by erg, for example by application of E-4031.

  1. In the results shown in Fig. 4 to 8, controls are missing.

  1. The sense behind application of serotonin to GH3 cells remains unclear. Do GH3 cells have serotonin receptors?

Minor points:

  1. The model GH3 cell is unsuitable for investigating the antidepressant effects of VOR since the neuronal target of VOR is unknown. From the results presented here, it remains unclear which distinct ion channel type is affected by VOX and if such channel is expressed in the neuronal targets of the antidepressant VOR. With the experiments presented here, only another drug inhibiting Kv channels in GH3 cells is shown. This should be discussed.

Author Response

Comments and Suggestions for Authors

The authors conducted sophisticated experiments to describe the effects of the antidepressant drug Vortixetine (VOR) on ionic currents in rat pituitary GH3 cells. According to their results, they conclude that VOR suppressed delayed-rectifier K+ currents (IKdr) whereby VOR differentially inhibited the peak and sustained components of this current. Furthermore, they conclude that the modulation of currents by VOR conceivably represents a part of the mechanisms through which it possessed pharmacological properties and functional influence in various neurological disorders (e.g., major depression).

Ans: Thanks for the comments raised by the reviewer.

I have several major problems with the presented data and the conclusions drawn.

  1. The main issue is the type of current inhibited by VOR and the resulting effect on the current kinetics. GH3 cells express not only delayed rectifier but also inactivating K+ channels (IKA type). If VOR inhibits selectively IKdr, the remaining IKA current will change the kinetics of the activated currents and this is not caused by an influence of VOR on IKdr kinetics. Therefore, the authors must show that their currents are mediated by IKdr only, for example by application of 4-AP. The same applies to the experiments on Neuro-2a cells.

Ans: Thanks for the comment pointed out by the reviewer. In an effort to diminish the concerns raised by the reviewer, an additional set of experiments regarding the effect of 4-aminopyridine (4-AP) plus VOR on IK(DR) in Neuro-2a cells were further performed. Hence, the experimental results were incorporated into the revised manuscript. That is, “Furthermore, as cells were continually exposed to 4-aminopyrine (5 mM), VOR-mediated modification of IK(DR) amplitude in these cells remained effective. 4-Aminopyridine was shown to be an inhibitor of A-type K+ current. Therefore, like VOR actions on IK(DR) in GH3 cells, VOR is effective at suppressing IK(DR) in Neuro-2a cells.” (lines 518-520 in the revised manuscript).

  1. The authors should show to which extend their VOR-blocked IKdr current is mediated by erg, for example by application of E-4031.

Ans: As per the reviewer’s advice, another set of measurements was hence performed with respect to effect of VOR plus E-4031 on IK(DR) in GH3 cells. The results were also included in the revised manuscript. That is, “Moreover, in the continued presence of VOR (3 mM), a subsequent addition of E-4031 (10 mM), an inhibitor of erg-mediated K+ current (IK(erg)), did not inhibit current amplitude further (698 ± 37 pA [in the presence of VOR] versus 697 ± 38 pA [in the presence of VOR plus E-4031]; n = 7, P > 0.05).” (lines 226-229 in the revised manuscript). The source of 4-aminopyridine and E-4031 was also included in the Materials and Methods section of the revised manuscript (line 96).

  1. In the results shown in Fig. 4 to 8, controls are missing.

Ans: Thanks for the reviewer’s comments. The main reason why lack of graphs were shown in controls is that the biophysical properties of IK(DR) evoked in response to the long-lasting step depolarization were characterized by small current inactivation and fast deactivation, as demonstrated in Figure 3A of the manuscript. Therefore, we were unable to precisely observe either the quasi-steady-state inactivation curve, the cumulative inactivation, or the recovery of IK(DR) inactivation in the control cells. However, under cell exposure to VOR, IK(DR) was consistently demonstrated to decrease in concurrence with an increase in the inactivation time course of the current. As such, the steady-state inactivation curve, cumulative inactivation, or the recovery of current inactivation of IK(DR) acquired in the presence of 10 and 30 mM VOR could be demonstrated. The experimental results of ours showed the ability of VOR to produce a concentration-dependent change in the inactivation curve, cumulative inactivation, and the recovery of current inactivation of IK(DR), though no results in the controls could be demonstrated.

  1. The sense behind application of serotonin to GH3 cells remains unclear. Do GH3 cells have serotonin receptors?

Ans: Thanks for the reviewer’s comments. Assuming that VOR can suppress serotonin (5-HT) reuptake in these cells, one would expect that the extracellular concentration of serotonin could have been elevated. However, under our experimental conditions, as GH3 cells were exposed to serotonin (10 mM) alone, the IK(DR) amplitude remained unchanged. It is hence reasonable to assume that VOR-mediated change in the amplitude and gating of IK(DR) demonstrated here is most unlikely to be linked to its inhibition of serotonin reuptake.

Minor points:

  1. The model GH3 cell is unsuitable for investigating the antidepressant effects of VOR since the neuronal target of VOR is unknown. From the results presented here, it remains unclear which distinct ion channel type is affected by VOX and if such channel is expressed in the neuronal targets of the antidepressant VOR. With the experiments presented here, only another drug inhibiting Kv channels in GH3 cells is shown. This should be discussed.

Ans: Thanks for the reviewer’s comment. An additional paragraph related to this issue was hence included in the Discussion section of the revised manuscript. That is, “It needs to be stressed that the model GH3 cells used in this study might not be suitable for investigation the antidepressant effects of VOR, since the neuronal target of this drug is incompletely understood. Therefore, it still remains unclear which distinct types of ionic channels can be affected by VOR or whether such channels involved are expressed in the neuronal targets of the antidepressant VOR.” (lines 607-611 in the revised manuscript).

Submission Date         14 February 2022

Date of this review      07 Mar 2022 10:57:34

Reviewer 2 Report

In this manuscript, authors wanted to demonstrate the inhibitory effect of Vortioxetine on the magnitude and gating of delayed rectifier potassium currents. Although the authors conducted a series of electrophysiological experiments in vitro, I don't think they provided enough evidences to demonstrate the effect is independent of the binding to serotonin receptors or inhibition of serotonin reuptake. The conclusion should become more convincible if authors provides more evidences in in vivo conditions, including the electrophysiology and the serotonin concentration measurement in CSF.

Author Response

Comments and Suggestions for Authors

In this manuscript, authors wanted to demonstrate the inhibitory effect of Vortioxetine on the magnitude and gating of delayed rectifier potassium currents. Although the authors conducted a series of electrophysiological experiments in vitro, I don't think they provided enough evidences to demonstrate the effect is independent of the binding to serotonin receptors or inhibition of serotonin reuptake. The conclusion should become more convincible if authors provides more evidences in in vivo conditions, including the electrophysiology and the serotonin concentration measurement in CSF.

Ans: Thanks for the comments pointed out by the reviewer. A series of electrophysiological measurements has been indeed extensively made in the present investigations. However, in order to diminish the concerns raised by the reviewer, the text in the revised manuscript was revised and modified. An additional sentence was included at the end of the Discussion section of the revised manuscript. That is, “Noticeably, additional evidence in the in vivo conditions, such as the electrophysiological recording and the serotonin concentration measurement in cerebrospinal fluid, still needs to be tested to support the notion that VOR-induced changes in K+ currents are independent of its binding to serotonin receptors.” (lines 602-606 in the revised manuscript).

Submission Date       14 February 2022

Date of this review    31 Mar 2022 04:14:40

Reviewer 3 Report

The authors presented a valuable study regarding the effects of the inhibition and gating potassium channels. The paper is well designed and presented. I would suggest the Authors highlight the aim of the study in the Abstract and include in the final conclusions how the obtained findings can be implemented or usable in the case of neurodegenerative disorders. 

Author Response

Comments and Suggestions for Authors

The authors presented a valuable study regarding the effects of the inhibition and gating potassium channels. The paper is well designed and presented. I would suggest the Authors highlight the aim of the study in the Abstract and include in the final conclusions how the obtained findings can be implemented or usable in the case of neurodegenerative disorders. 

Ans: Thanks for the reviewer’s comments regarding the manuscript of ours on ways to improve the present manuscript. The aim regarding this study was included in the Abstract section of the revised manuscript. That is, “The aim in the current study was to explore possible modifications of VOR and other related compounds on ionic currents in pituitary GH3 cells and in Neuro-2a cells” (lines 16-17 in the revised manuscript). Moreover, at the end of the first paragraph shown in the Discussion section of the revised manuscript, the text was appropriately rephrased. That is, “Together with these data, the modulation of transmembrane ionic currents exerted by VOR conceivably represents a part of the fundamentally molecular mechanisms through which it possessed pharmacological properties and functional influence in various neurological disorders (e.g., major depression and other neurodegenerative disorders).” (lines 544-548 in the revised manuscript).

Submission Date       14 February 2022

Date of this review    13 Apr 2022 09:51:26

Reviewer 4 Report

The authors investigate the effects of Vortixetine (VOR) on delayed-rectifier K+ current (IK(DR)) and found VOR suppressed the amplitude of IK(DR) in a concentration-, time-, and state-dependent manner. They found the same phenotype in different cells. Overall, the experiments are well-conducted, and the methodological approach is appropriate. I do not have major concerns about this study; however, I have a number of minor criticisms that need to be addressed by the authors.

  1. Although VOR can inhibit IK(DR), it will be interesting to see whether it can inhibit other types of K+ channels and maybe Na+ or Ca2+ Please add more experiments to clarify that.
  2. The authors only test two cell lines, it is necessary to test the effect of VOR on cultured primary neurons.
  3. The is a typo in Figure6, it should be μM not mM.

Author Response

Comments and Suggestions for Authors

The authors investigate the effects of Vortixetine (VOR) on delayed-rectifier K+ current (IK(DR)) and found VOR suppressed the amplitude of IK(DR) in a concentration-, time-, and state-dependent manner. They found the same phenotype in different cells. Overall, the experiments are well-conducted, and the methodological approach is appropriate. I do not have major concerns about this study; however, I have a number of minor criticisms that need to be addressed by the authors.

Ans: We are grateful for the comments raised by the reviewer regarding the manuscript of ours.

  1. Although VOR can inhibit IK(DR), it will be interesting to see whether it can inhibit other types of K+ channels and maybe Na+ or Ca2+ Please add more experiments to clarify that.

Ans: Thanks for the comments provided by the reviewer. Additional experiments regarding effect of VOR on voltage-gated Na+ or L-type Ca2+ currents in GH3 cells were performed.  Hence, the text relevant to this issue was included in the revised manuscript (lines 542-544). The preparation of electrode used for measurement of Na+ or Ca2+ currents was also included in the revised manuscript (lines 111-113).

  1. The authors only test two cell lines, it is necessary to test the effect of VOR on cultured primary neurons.

Ans: Thanks for the reviewer’s comment. Since the total number of Figures shown in this manuscript has been reached to 10, the space of the manuscript used for inclusion of VOR effect on cultured primary neurons could become enlarged. The issue was hence included in the Discussion section of the revised manuscript. That is, “However, whether or how VOR-mediated changes in IK(DR) could occur in cultured primary neurons remains to be further studied.” (lines 581-582)

  1. The is a typo in Figure6, it should be μM not mM.

Ans: Thanks to the reviewer for bringing our attention. Sorry! We made a mistake. “mM” was hence replaced with “microM” (line 406 in the revised manuscript).

Submission Date       14 February 2022

Date of this review    17 Apr 2022 05:33:00

Round 2

Reviewer 1 Report

  1. The main issue is the type of current inhibited by VOR and the resulting effect on the current kinetics. GH3 cells express not only delayed rectifier but also inactivating K+ channels (IKA type). If VOR inhibits selectively IKdr, the remaining IKA current will change the kinetics of the activated currents and this is not caused by an influence of VOR on IKdr kinetics. Therefore, the authors must show that their currents are mediated by IKdr only, for example by application of 4-AP. The same applies to the experiments on Neuro-2a cells.

Ans: Thanks for the comment pointed out by the reviewer. In an effort to diminish the concerns raised by the reviewer, an additional set of experiments regarding the effect of 4-aminopyridine (4-AP) plus VOR on IK(DR) in Neuro-2a cells were further performed. Hence, the experimental results were incorporated into the revised manuscript. That is, “Furthermore, as cells were continually exposed to 4-aminopyrine (5 mM), VOR-mediated modification of IK(DR) amplitude in these cells remained effective. 4-Aminopyridine was shown to be an inhibitor of A-type K+ current. Therefore, like VOR actions on IK(DR) in GH3 cells, VOR is effective at suppressing IK(DR) in Neuro-2a cells.” (lines 518-520 in the revised manuscript).

Data are not provided. Experiments on GH3 cells were not performed. Examples of the effect of 4-AP on currents with and without VOR must be shown.

  1. The authors should show to which extend their VOR-blocked IKdr current is mediated by erg, for example by application of E-4031.

Ans: As per the reviewer’s advice, another set of measurements was hence performed with respect to effect of VOR plus E-4031 on IK(DR) in GH3 cells. The results were also included in the revised manuscript. That is, “Moreover, in the continued presence of VOR (3 µM), a subsequent addition of E-4031 (10 µM), an inhibitor of erg-mediated K+ current (IK(erg)), did not inhibit current amplitude further (698 ± 37 pA [in the presence of VOR] versus 697 ± 38 pA [in the presence of VOR plus E-4031]; n = 7, P > 0.05).” (lines 226-229 in the revised manuscript). The source of 4-aminopyridine and E-4031 was also included in the Materials and Methods section of the revised manuscript (line 96).

Effect of E-4031 must also be shown in Neuro-2a cells.

  1. In the results shown in Fig. 4 to 8, controls are missing.

Ans: Thanks for the reviewer’s comments. The main reason why lack of graphs were shown in controls is that the biophysical properties of IK(DR) evoked in response to the long-lasting stepdepolarization were characterized by small current inactivation and fast deactivation, as demonstrated in Figure 3A of the manuscript. Therefore, we were unable to precisely observe either the quasi-steady-state inactivation curve, the cumulative inactivation, or the recovery of IK(DR) inactivation in the control cells. However, under cell exposure to VOR, IK(DR) was consistently demonstrated to decrease in concurrence with an increase in the inactivation time course of the current. As such, the steady-state inactivation curve, cumulative inactivation, or the recovery of current inactivation of IK(DR) acquired in the presence of 10 and 30 µM VOR could be demonstrated. The experimental results of ours showed the ability of VOR to produce a concentration-dependent change in the inactivation curve, cumulative inactivation, and the recovery of current inactivation of IK(DR), though no results in the controls could be demonstrated.

This is completely unclear for me. If controls in the corresponding figures could not be shown, then the currents shown under exposure to VOR do not display IK(DR) currents but rather currents not blocked by VOR, for example IKA. This would contradict the statement that VOR influences the kinetics of IK(DR).

Author Response

English language and style

( ) Extensive editing of English language and style required
( ) Moderate English changes required
(x) English language and style are fine/minor spell check required
( ) I don't feel qualified to judge about the English language and style

Yes

Can be improved

Must be improved

Not applicable

Does the introduction provide sufficient background and include all relevant references?

(x)

( )

( )

( )

Is the research design appropriate?

( )

( )

(x)

( )

Are the methods adequately described?

(x)

( )

( )

( )

Are the results clearly presented?

( )

( )

(x)

( )

Are the conclusions supported by the results?

( )

( )

(x)

( )

Comments and Suggestions for Authors

  1. The main issue is the type of current inhibited by VOR and the resulting effect on the current kinetics. GH3 cells express not only delayed rectifier but also inactivating K+ channels (IKA type). If VOR inhibits selectively IKdr, the remaining IKA current will change the kinetics of the activated currents and this is not caused by an influence of VOR on IKdr kinetics. Therefore, the authors must show that their currents are mediated by IKdr only, for example by application of 4-AP. The same applies to the experiments on Neuro-2a cells.

Ans: Thanks for the comment pointed out by the reviewer. In an effort to diminish the concerns raised by the reviewer, an additional set of experiments regarding the effect of 4-aminopyridine (4-AP) plus VOR on IK(DR) in Neuro-2a cells were further performed. Hence, the experimental results were incorporated into the revised manuscript. That is, “Furthermore, as cells were continually exposed to 4-aminopyrine (5 mM), VOR-mediated modification of IK(DR) amplitude in these cells remained effective. 4-Aminopyridine was shown to be an inhibitor of A-type K+ current. Therefore, like VOR actions on IK(DR) in GH3 cells, VOR is effective at suppressing IK(DR) in Neuro-2a cells.” (lines 518-520 in the revised manuscript).

Data are not provided. Experiments on GH3 cells were not performed. Examples of the effect of 4-AP on currents with and without VOR must be shown.

Ans: As advised by the reviewer, the results showing effect of 4-AP on currents with and without VOR were shown in the revised manuscript. That is, “As cells were continually exposed to 4-aminopyridine (4-AP, 5 mM), VOR-mediated modifications on IK(DR) in GH3 cells remained effective. For example, in continued presence of 5 mM 4-AP, the IK(DR) at the end of depolarizing pulse was significantly decreased from 644 ± 31 to 508 ± 25 pA (n = 7, P < 0.05) during cell exposure to 3 mM VOR. 4-AP was reported to be an inhibitor of A-type K+ currents.” (lines 230-235 of the revised manuscript).

  1. The authors should show to which extend their VOR-blocked IKdr current is mediated by erg, for example by application of E-4031.

Ans: As per the reviewer’s advice, another set of measurements was hence performed with respect to effect of VOR plus E-4031 on IK(DR) in GH3 cells. The results were also included in the revised manuscript. That is, “Moreover, in the continued presence of VOR (3 µM), a subsequent addition of E-4031 (10 µM), an inhibitor of erg-mediated K+ current (IK(erg)), did not inhibit current amplitude further (698 ± 37 pA [in the presence of VOR] versus 697 ± 38 pA [in the presence of VOR plus E-4031]; n = 7, P > 0.05).” (lines 226-229 in the revised manuscript). The source of 4-aminopyridine and E-4031 was also included in the Materials and Methods section of the revised manuscript (line 96).

Effect of E-4031 must also be shown in Neuro-2a cells.

Ans: Thanks for the comments provided by the reviewer. The results were included in the revised manuscript. That is, “Furthermore, as cells were continually exposed to 4-AP (5 mM) or E-4031 (10 mM), VOR-mediated modification of IK(DR) amplitude seen in these cells remained effective.” (lines 525-527 in the revised manuscript).

  1. In the results shown in Fig. 4 to 8, controls are missing.

Ans: Thanks for the reviewer’s comments. The main reason why lack of graphs were shown in controls is that the biophysical properties of IK(DR) evoked in response to the long-lasting stepdepolarization were characterized by small current inactivation and fast deactivation, as demonstrated in Figure 3A of the manuscript. Therefore, we were unable to precisely observe either the quasi-steady-state inactivation curve, the cumulative inactivation, or the recovery of IK(DR) inactivation in the control cells. However, under cell exposure to VOR, IK(DR) was consistently demonstrated to decrease in concurrence with an increase in the inactivation time course of the current. As such, the steady-state inactivation curve, cumulative inactivation, or the recovery of current inactivation of IK(DR) acquired in the presence of 10 and 30 µM VOR could be demonstrated. The experimental results of ours showed the ability of VOR to produce a concentration-dependent change in the inactivation curve, cumulative inactivation, and the recovery of current inactivation of IK(DR), though no results in the controls could be demonstrated.

This is completely unclear for me. If controls in the corresponding figures could not be shown, then the currents shown under exposure to VOR do not display IK(DR) currents but rather currents not blocked by VOR, for example IKA. This would contradict the statement that VOR influences the kinetics of IK(DR).

Ans: Thanks for the insightful comments. To minimize the concern raised by the reviewer, the control data were included in the revised manuscript. Hence, Figures 4A and 4B were redone in attempts to show current traces obtained in the control period. The text in the figure legend was accordingly modified in the revised version of the manuscript.

Submission Date       14 February 2022

Date of this review    03 May 2022 12:42:47

Reviewer 2 Report

I don't think the authors solved my concerns and I do not believe the manuscript has been sufficiently improved to warrant publication in Biomedicines.

Author Response

English language and style

( ) Extensive editing of English language and style required
(x) Moderate English changes required
( ) English language and style are fine/minor spell check required
( ) I don't feel qualified to judge about the English language and style

Yes

Can be improved

Must be improved

Not applicable

Does the introduction provide sufficient background and include all relevant references?

( )

(x)

( )

( )

Is the research design appropriate?

( )

(x)

( )

( )

Are the methods adequately described?

( )

(x)

( )

( )

Are the results clearly presented?

( )

(x)

( )

( )

Are the conclusions supported by the results?

( )

(x)

( )

( )

Comments and Suggestions for Authors

I don't think the authors solved my concerns and I do not believe the manuscript has been sufficiently improved to warrant publication in Biomedicines.

Ans: Thanks for the comments. To minimize the concerns, the revised version of the manuscript has been carefully revised and modified. Figure 4 in the revised manuscript has been revised and the figure legend was appropriately modified (lines 342-351 and lines 358-364). Additional results (lines 230-235 and lines 525-527) were also included in the manuscript. It is sincerely hoped that the manuscript of ours will satisfy the reviewer’s comments.

Submission Date       14 February 2022

Date of this review    29 Apr 2022 04:21:26

Round 3

Reviewer 1 Report

Examples of the effect of 4-AP on currents with and without VOR must be shown.

Tis is still an open demand. To judge the effect of VOR on current kinetics, examples of the effect of 4-AP on currents with and without VOR must be shown.

Effect of E-4031 must also be shown in Neuro-2a cells.

This is also not fulfilled by the authors. Data are necessary.

  1. In the results shown in Fig. 4 to 8, controls are missing.

The authors did provide controls for Fig. 4 only, not for the others.

Author Response

English language and style

( ) Extensive editing of English language and style required
( ) Moderate English changes required
(x) English language and style are fine/minor spell check required
( ) I don't feel qualified to judge about the English language and style

Yes

Can be improved

Must be improved

Not applicable

Does the introduction provide sufficient background and include all relevant references?

(x)

( )

( )

( )

Is the research design appropriate?

( )

( )

(x)

( )

Are the methods adequately described?

( )

(x)

( )

( )

Are the results clearly presented?

( )

( )

(x)

( )

Are the conclusions supported by the results?

( )

( )

(x)

( )

Comments and Suggestions for Authors

Examples of the effect of 4-AP on currents with and without VOR must be shown.

This is still an open demand. To judge the effect of VOR on current kinetics, examples of the effect of 4-AP on currents with and without VOR must be shown.

Ans: As per the comments provided by the reviewer, the experimental results were accordingly included in this version of the manuscript. That is, “Additionally, in continued presence of 10 mM VOR, further addition of 5 mM 4-AP or E-4031 (10 mM) did not alter IK(DR) amplitude further in Neuro-2a cells (285 ± 17 pA [in the presence of VOR], 284 ± 18 pA [in the presence of VOR plus 4-AP], versus 283 ± 18 pA [in the presence of VOR plus E-4031]; n = 7, P > 0.05).” (lines 523-526 in this version of revised manuscript).

 Effect of E-4031 must also be shown in Neuro-2a cells.

Ans: Thanks for the reviewer’s comments. The data were incorporated into the revised version of the manuscript. That is, “Additionally, in continued presence of 10 mM VOR, further addition of 5 mM 4-AP or E-4031 (10 mM) did not alter IK(DR) amplitude further in Neuro-2a cells (285 ± 17 pA [in the presence of VOR], 284 ± 18 pA [in the presence of VOR plus 4-AP], versus 286 ± 19 pA [in the presence of VOR plus E-4031]; n = 7, P > 0.05).” (lines 523-526 in this version of revised manuscript).

 This is also not fulfilled by the authors. Data are necessary.

 In the results shown in Fig. 4 to 8, controls are missing.

 The authors did provide controls for Fig. 4 only, not for the others.

Ans: Thanks for the reviewer’s comment. Figure 6B was hence redone because the control data were included in the text of the revised manuscript. The text in the manuscript was accordingly and appropriately rephrased. However, as demonstrated in Figure 4A, because current traces with inactivating properties appeared to comparatively small in the control period (i.e., absence of VOR), Figures 5, 7 and 8 in this version of the revised manuscript thus remained unchanged.

Ans: Thanks for the comments.

Reviewer 2 Report

The manuscript was much improved after several revisons. 

Author Response

English language and style

( ) Extensive editing of English language and style required
( ) Moderate English changes required
( ) English language and style are fine/minor spell check required
(x) I don't feel qualified to judge about the English language and style

Yes

Can be improved

Must be improved

Not applicable

Does the introduction provide sufficient background and include all relevant references?

(x)

( )

( )

( )

Is the research design appropriate?

( )

(x)

( )

( )

Are the methods adequately described?

(x)

( )

( )

( )

Are the results clearly presented?

(x)

( )

( )

( )

Are the conclusions supported by the results?

(x)

( )

( )

( )

Comments and Suggestions for Authors

The manuscript was much improved after several revisions. 

Ans: Thanks for the comments provided by the reviewer.